# The Promise of Circulating Tumor DNA (ctDNA) in the Management of Early-Stage Colon Cancer: A Critical Review

**DOI:** 10.3390/cancers12102808

**Published:** 2020-09-29

**Authors:** Sakti Chakrabarti, Hao Xie, Raul Urrutia, Amit Mahipal

**Affiliations:** 1Department of Hematology-Oncology, Medical College of Wisconsin, 8701 Watertown Plank Road, Milwaukee, WI 53226, USA; 2Department of Gastrointestinal Oncology, Moffitt Cancer Center, 12902 USF Magnolia Drive, Tampa, FL 33612, USA; Hao.Xie@moffitt.org; 3Department of Surgery, Medical College of Wisconsin, 8701 Watertown Plank Road, Milwaukee, WI 53226, USA; rurrutia@mcw.edu; 4Division of Medical Oncology, Mayo Clinic, 200 First Street SW, Rochester, MN 55905, USA; mahipal.amit@mayo.edu

**Keywords:** circulating tumor DNA (ctDNA), colon cancer, adjuvant therapy, next-generation sequencing, minimal residual disease (MRD), early-stage colon cancer

## Abstract

**Simple Summary:**

Currently, the treatment for localized colon cancer consists of surgery and, if the presence of residual cancer cells is suspected, chemotherapy following the surgery. However, the current standard tools to determine the presence of residual cancer after the surgery are imprecise, which results in under- or overtreatment in a significant number of patients. Emerging research indicates that circulating tumor DNA (ctDNA) can reveal the presence of residual cancer after surgery with much higher precision than the presently available tools, and can help with the treatment decision regarding a need for chemotherapy after the surgery. Furthermore, ctDNA can potentially help determine the effectiveness of chemotherapy and detect cancer recurrence much earlier than the current standard tools. In this review, we have critically evaluated available data to provide the readers with an overview of how ctDNA can potentially transform the treatment of localized colon cancer in the near future.

**Abstract:**

The current standard treatment for patients with early-stage colon cancer consists of surgical resection, followed by adjuvant therapy in a select group of patients deemed at risk of cancer recurrence. The decision to administer adjuvant therapy, intended to eradicate the clinically inapparent minimal residual disease (MRD) to achieve a cure, is guided by clinicopathologic characteristics of the tumor. However, the risk stratification based on clinicopathologic characteristics is imprecise and results in under or overtreatment in a substantial number of patients. Emerging research indicates that the circulating tumor DNA (ctDNA), a fraction of cell-free DNA (cfDNA) in the bloodstream that originates from the neoplastic cells and carry tumor-specific genomic alterations, is a promising surrogate marker of MRD. Several recent studies suggest that ctDNA-guided risk stratification for adjuvant therapy outperforms existing clinicopathologic prognostic indicators. Preliminary data also indicate that, aside from being a prognostic indicator, ctDNA can inform on the efficacy of adjuvant therapy, which is the underlying scientific rationale for several ongoing clinical trials evaluating ctDNA-guided therapy escalation or de-escalation. Furthermore, serial monitoring of ctDNA after completion of definitive therapy can potentially detect cancer recurrence much earlier than conventional surveillance methods that may provide a critical window of opportunity for additional curative-intent therapeutic interventions. This article presents a critical overview of published studies that evaluated the clinical utility of ctDNA in the management of patients with early-stage colon cancer, and the potential of ctDNA to transform the adjuvant therapy strategies.

## 1. Introduction

Colorectal cancer (CRC) is the third most commonly diagnosed cancer and the second leading cause of cancer death worldwide [1]. In the United States, CRC ranks second as the cause of cancer-related death, with 148,000 new cases diagnosed annually [2]. However, approximately 80% of newly diagnosed CRC patients present with early-stage disease, allowing an opportunity for curative-intent treatment [2]. Early-stage colon cancer (ESCC) refers to stages I, II, and III in the 8th American Joint Committee on Cancer (AJCC) staging system, and encompasses tumors confined to the colon and the adjacent structures, with or without involvement of the regional lymph nodes [3]. Current standard treatment of early-stage colon cancer consists of surgical resection of the primary colonic tumor along with the regional lymph nodes, and adjuvant chemotherapy (ACT) in a select group of patients deemed at risk of cancer recurrence despite surgery [4]. The goal of ACT is a cure by eradicating clinically inapparent micrometastatic disease, also known as minimal residual disease (MRD). One of the most enduring challenges in the adjuvant therapy paradigm of colon cancer is the lack of a reliable biomarker that strongly correlates with the presence of MRD and helps precise risk stratification of patients for adjuvant therapy. In the current clinical practice, patient selection for ACT is based on clinicopathologic characteristics of the tumor, which is imprecise and leads to under or overtreatment in a substantial number of patients [5,6], underscoring an overwhelming need of a reliable surrogate marker for MRD assessment.

Emerging research suggests that the circulating tumor DNA (ctDNA), a component of cell-free DNA (cfDNA) that carries tumor-specific genomic or epigenomic alterations, may serve as a surrogate marker of MRD and help risk-stratify resected early-stage colon cancer patients with a high degree of precision [7,8,9,10]. Congruently, several studies have reported that serial monitoring of ctDNA can provide valuable information on the efficacy of ACT, and detect cancer recurrence much earlier than standard surveillance methods [7,8,9,10]. In this article, we summarize the most important aspects of the ctDNA biology, discuss current paradigms that support the clinical utility of ctDNA in the detection and monitoring of MRD, in the assessment of the efficacy of ACT, and early detection of cancer recurrence. Furthermore, we describe the rationale and design of the ongoing clinical trials investigating the validity of ctDNA-guided strategies in the management of early-stage colon cancer.

## 2. The Biology of ctDNA

ctDNA is single- or double-stranded DNA fragments released from neoplastic cells, which typically constitute less than 1% of the total cfDNA [11]. Pioneering investigation by Leon et al. back in the 1970s led to the realization that the rapid cell turnover rate in malignant tumors results in an increased concentration of cfDNA in the blood of cancer patients compared to healthy individuals [12], which was subsequently confirmed by other investigators [13]. Noteworthy, however, several conditions unrelated to cancer, such as acute trauma, ischemia, infection, or inflammation, can increase cfDNA concentrations in the circulation [14,15]. ctDNA is released into the circulation predominantly by apoptosis [16], necrosis, phagocytosis, and carried by exosomes [17,18]. ctDNA can also be detected in non-blood body fluids, such as urine, saliva, sputum, stool, pleural fluid, and cerebrospinal fluid (CSF) [19]. These DNA fragments are continuously released by neoplastic cells, undergo rapid degradation by blood nucleases, and are finally cleared by the liver and kidneys [20], accounting for their short half-life in circulation (16 min to 2.5 h) [21]. The rapid turnover of ctDNA in circulation makes it an attractive target for obtaining a real-time account of mutation dynamics and tumor burden [22,23]. Additionally, DNA fragments derived from cancer cells are typically shorter in length, which forms the basis for selecting fragments between 90 bp and 150 bp to improve the detection sensitivity of ctDNA assays [24]. Notably, plasma samples are preferable to serum samples for ctDNA analysis as the latter contain larger quantities of DNA from leukocytes lysed during the clotting process, and thereby, increasing the background vs. signals ratio and interfering with the assay [25].

## 3. Methodological Considerations for the Use of ctDNA in Colon Cancer

The probability of detecting ctDNA in plasma depends on the tumor burden. For colorectal cancer, the rate of ctDNA detection ranges from 73% in localized disease to nearly 100% in metastatic disease [26]. Furthermore, the ctDNA detection rate drops significantly following curative resection, ranging from 10%–15% in patients with stage II disease to nearly 50% in those with stage IV disease [9,10,26,27,28]. In patients with early-stage colon cancer following curative surgery, the ctDNA fraction of the total cell-free DNA is often less than 0.1% [10,29]. Therefore, methods for obtaining circulating DNA with high yield and consistency, as well as analytical platforms with high sensitivity, are necessary to measure ctDNA in the plasma for effective MRD assessment.

Contemporary ctDNA assays can be broadly divided into two major categories—polymerase chain reaction (PCR)-based assays and next-generation sequencing (NGS)-based assays. Droplet digital PCR (ddPCR) [30] is a prime example of the former category that has been utilized by several groups [8,31]. In ddPCR, the plasma sample is partitioned using a droplet generator into numerous discrete droplets such that each droplet contains no more than one fragment of the template DNA. DNA fragment in each droplet is then analyzed simultaneously for target sequences through an endpoint PCR, allowing detection of the mutations of interest. Another example of PCR-based assay is beads, emulsion, amplification, and magnetics (BEAMing), which has not gained similar popularity as ddPCR because of the complexity of the procedure [32]. Moreover, ddPCR-based assays are inexpensive, fast, and have high sensitivity with a variant allele frequency (VAF) for detection of ≤ 0.01% [33]. However, this type of assay is limited to detection of a small number of mutations, and unable to detect mutations not known a priori, thus limiting its ability to address the issues related to intratumor heterogeneity and emergent mutations [34].

In contrast, NGS-based assays can assess mutations across broad areas of the genome, and are not limited to testing for known mutations, a distinct advantage over ddPCR [21]. Among the NGS-based assays, targeted sequencing platforms, including tagged-amplicon deep sequencing (TAm-Seq) [35], safe-sequencing system (Safe-SeqS) [36], and cancer personalized profiling by deep sequencing (CAPP-Seq) [37], are the popular novel platforms. These assays have detection limits as low as 0.01%, but with the downside of being more expensive and time-consuming. Whole-exome (WES) [38] and whole-genome (WGS) [39] sequencing, the other less popular NGS-based platforms, are limited by sensitivity and cost.

The current paradigm divides the assays described above as tumor informed or tumor uninformed. Briefly, tumor-informed assays rely on data derived from NexGen sequencing of tumor tissues to select target mutations for testing [7,8,9,27,31]. Accordingly, primers are designed against the specific genomic targets identified, which helps increase the depth, improve sensitivity, and reduce the probability of false-positive results related to sequencing errors. This approach substantially reduces false-positive results secondary to clonal hematopoiesis of indeterminate potential (CHIP), which might otherwise be misinterpreted as tumor-derived DNA [40]. However, sequencing of the tumor incurs additional expenses, and more importantly, may cause reporting delay and hinder timely initiation of ACT, negatively impacting survival [41]. Furthermore, tumor sequencing might not detect all relevant mutations because of intratumor heterogeneity [42] and may not capture subsequently emerging mutations as a result of treatment [43]. Another relevant shortcoming of the tumor-informed assays is that sequencing errors might be indistinguishable from actual mutations, especially if the mutations have a low VAF (<0.01%). This issue, however, can be circumvented by using molecular barcoding, in which each molecule in a sample is tagged with a unique molecular barcode enabling the sequence analysis software filter out duplicate reads and PCR errors to report unique reads [33,44]. Figure 1 illustrates the sensitivity levels of common ctDNA assays.

## 4. Role of ctDNA in the Detection of Minimal Residual Disease (MRD)

Surgery alone can cure a vast majority of patients with early-stage colon cancer. A retrospective analysis of the Swedish Colorectal Cancer Registry data reported a 5-year disease-free survival (DFS) rate of 78% with surgery alone in the low-risk subgroup of stage III patients (patients with T1-T3, N1 disease, and no additional risk factors), and 78% to 91% of 5-year DFS rate in patients with stage II disease [45]. If all stage III patients are considered as a single group, nearly 50% of these patients can be cured by surgery alone [5]. However, ACT is recommended for all stage III patients in current treatment guidelines [4,41] because a biomarker does not exist at this time that can reliably identify patients with MRD who are truly at risk of cancer recurrence. MOSAIC trial reported a 5-year DFS rate of approximately 67% in stage III patients who received six months of oxaliplatin-based ACT [46] Based on these data, it is reasonable to infer that among the stage III patients who receive six months of oxaliplatin-based ACT, only 17% of patients derive a survival benefit from ACT. Moreover, this gain in survival with oxaliplatin-based ACT should be weighed against the short- and long-term treatment-related toxicities, especially 12.5% of grade 3 neuropathy after six months of treatment [46]. Therefore, finding a surrogate biomarker for the detection of MRD in resected early-stage colon cancer patients is of utmost importance. Table 1 summarizes the key studies that evaluated the feasibility of using ctDNA as a surrogate biomarker for MRD detection.

Diehl et al. reported a study more than a decade ago evaluating the feasibility of MRD detection through peripheral blood ctDNA testing in a group of advanced-stage colon cancer patients undergoing curative-intent resection [23]. In this study, ctDNA measurements were performed in 18 patients (16 out of 18 had stage IV disease) before and after the surgery. All but one patient with detectable postoperative ctDNA had cancer recurrence as opposed to zero out of four patients who had undetectable ctDNA after surgery. The difference in the recurrence rate between subjects with and without detectable ctDNA at the first postoperative follow-up was highly statistically significant (*p* = 0.006), underscoring the potential of this approach to detect MRD.

A series of subsequent studies provided evidences supporting the candidacy of ctDNA technology as a bona fide method of developing a biomarker for MRD. Using a tumor-informed Safe-SeqS platform-based ctDNA assay, Tie et al. reported two prospective, multicenter, cohort studies, one in stage II (*n* = 230) [10] and the other in stage III (*n* = 96) patients [9], showing that ctDNA significantly outperformed standard clinicopathologic characteristics as a prognostic marker. In their studies, tumor tissue was analyzed for somatic mutations in 15 genes commonly known to be mutated in CRC, and one mutation identified in the tumor tissue was selected for ctDNA testing in each patient. Among the patients in stage II cohort [10] who did not receive ACT (*n* = 178), 79% of the patients (11 out of 14) with detectable ctDNA postoperatively (4 to 10 weeks after surgery) had cancer recurrence at a median follow- up duration of 27 months (HR 18, 95% CI 7.9–40; *p* < 0.001). Conversely, only 9.8% (16 out of 164) of the patients with undetectable postoperative ctDNA had a cancer recurrence. On multivariable analysis, postoperative ctDNA status remained the strongest independent predictor of relapse-free survival (HR, 28; 95% CI, 11 to 68), and outperformed any individual clinicopathological risk factor or any combination of clinicopathological factors in predicting cancer recurrence. In the study with stage III patients [9], ctDNA was detectable in 20 out of 96 (21%) patients postoperatively (4–10 weeks after surgery) and the recurrence-free survival at 3 years in this group was 47% (95% CI, 24%–68%) compared to 76% in those with undetectable postoperative ctDNA (HR, 3.8; 95% CI, 2.4–21.0; *p* < 0.001). Similar to stage II patients, postoperative ctDNA status remained independently associated with recurrence-free interval after adjusting for known clinicopathologic risk factors (HR, 7.5; 95% CI, 3.5–16.1; *p* < 0.001).

A similar study was conducted by Reinert et al. in a cohort of 125 CRC patients (stages I to III) [7]. In this prospective, multicenter cohort study, ctDNA was quantified in the preoperative and postoperative plasma samples by a personalized tumor-informed, multiplex, polymerase chain reaction–based, next-generation sequencing platform. The study showed that the patients with detectable ctDNA at postoperative day 30 were seven times (HR, 7.2; 95% CI, 2.7–19.0; *p* < 0.001) more likely to have cancer recurrence compared to those with undetectable ctDNA. In multivariate analyses, ctDNA status was independently associated with recurrence after adjusting for known clinicopathologic risk factors.

A study by Tarazona et al. reported similar findings in which 150 patients with resected localized colon cancer underwent serial ctDNA testing for MRD evaluation [8]. This study utilized tumor-informed ddPCR-based ctDNA assay. Detection of ctDNA post-surgery strongly correlated with cancer relapse with HR of 17.56 (*p* = 0.0014). A retrospective analysis of the IDEA-France data also reported that the detection of ctDNA postoperatively is an independent adverse prognostic marker for cancer recurrence (adjusted HR,1.85; *p* < 0.001) in stage III patients treated with oxaliplatin-based ACT [49]. A similar study by Allegretti et al. [50] reported that persistence and absence of ctDNA at the time of first post-operative (3 month) follow-up were associated with fast relapse and a disease-free status in three and seven patients, respectively. In addition, this study reported improved sensitivity (58.8% to 63.6%) when ctDNA result was analyzed in combination with serum CEA level. Hence, when combined, these studies strongly support ctDNA as a potentially robust biomarker of MRD.

Despite remarkable heterogeneity among the reported studies in terms of pre-analytical variables, assay platforms, and outcomes measured, several conclusions can be drawn. First, ctDNA outperforms existing clinicopathologic risk factors as a prognostic biomarker [51] (Figure 2). Second, most patients with detectable ctDNA after completion of definitive therapy had a cancer recurrence suggesting a high degree of specificity of the ctDNA assays. A wide variety of factors, ranging from a short follow-up period to false-positivity of the assays, could be incriminated in a small number of patients in which post-therapy ctDNA detection did not correlate with cancer relapse. Finally, the ability of a single postoperative ctDNA test to predict disease relapse (sensitivity) is limited, approximately 50% [7,10]. Several clinical trials are currently underway to evaluate the validity of ctDNA as a surrogate biomarker of MRD in larger cohorts of patients (Table 2), including randomized phase II/III COBRA study (NCT0406810), the CIRCULATE trial (NCT04120701) and the DYNAMIC-II study (ACTRN12615000381583).

## 5. Role of ctDNA in Assessing the Efficacy of Adjuvant Therapy

Several studies have reported that a decrease in the ctDNA level during systemic therapy in metastatic CRC strongly correlates with tumor response [23,52,53,54], raising the question as to whether ctDNA can help inform clinicians on the efficacy of adjuvant therapy.

Prospective observational studies (summarized in Table 1) have reported a substantially lower risk of cancer recurrence if ctDNA detectable after surgery becomes undetectable after ACT [7,8,9,10]. In the study with stage II patients, ctDNA detection immediately after completion of ACT was associated with poorer RFS (HR, 11; 95% CI, 1.8 to 68; *p* = 0.001) [10]. In this study, two of six patients with detectable ctDNA after surgery had no detectable ctDNA after ACT and remained disease-free at a median follow-up of 27 months, whereas virtually all patients had recurrence if ctDNA was detectable after ACT. The study with stage III patients published by the same group [9] reported a 3-year recurrence-free survival of 30% if ctDNA was detectable after the completion of ACT compared to 77% if ctDNA was undetectable (HR, 6.8; 95% CI, 11.0–157.0; *p* < 0.001). In this cohort, ctDNA detectable after surgery became undetectable in 9 out of 18 patients after the completion of ACT, and was associated with an improved relapse-free survival relative to those who had detectable ctDNA after ACT (HR 5.1; *p* = 0.02) [55]. Reinert et al. reported a 17 times higher risk of recurrence (HR, 17.5; 95% CI, 5.4–56.5; *p* < 0 .001) if ctDNA remained detectable after completion of ACT [7]. Further analysis of the data revealed that 30% (3 out of 10) of patients in this cohort with detectable ctDNA postoperatively cleared ctDNA with ACT and were disease-free long term. The other 7 patients with persistently detectable ctDNA after ACT had disease relapse. Tarazona et al. reported an 85.7% recurrence rate among patients with detectable ctDNA post-ACT (HR 10.02; 95% CI 9.202−307.3; *p* < 0.0001). In this cohort, one out of seven patients cleared ctDNA with ACT and remained disease-free long term. Although the interpretation of these data is limited by small sample sizes and heterogeneous ctDNA assay platforms, these studies provide early evidence supporting the utility of ctDNA to inform on the efficacy of adjuvant therapy. However, clinical trials with larger cohorts of patients are needed before clearance of ctDNA with ACT is considered an acceptable surrogate marker of survival and adjuvant therapy effectiveness. Several large trials are currently underway to address this question (COBRA, CIRCULATE, and DYNAMIC-II) (Table 2). We are optimistic that the results of these trials will provide further guidance on this issue.

## 6. Potential Role of ctDNA in Surveillance

The purpose of surveillance after definitive therapy of colon cancer is early identification of cancer recurrence that might allow further curative-intent treatment. Approximately 5% to 30% of patients with early-stage colon cancer, depending on the stage at diagnosis, experience recurrence following the curative-intent therapy [41]. Current expert guidelines endorse intensive postoperative surveillance for patients with resected stage II and III colon cancer who would be considered candidates for curative-intent surgery [4]. However, the intensive surveillance protocol with periodic serum carcinoembryonic antigen (CEA) test, radiologic studies, and colonoscopy detects most recurrences late, allowing potentially curative treatment only in 10% to 20% of patients [56,57]. Furthermore, a recently published Cochrane analysis revealed that intensive surveillance led to more frequent salvage surgeries with a curative intent (risk ratio 1.98; 95% CI, 1.53–2.56), but did not appear to translate into a survival advantage [58]. In this context, several studies have reported exciting early data suggesting that ctDNA can diagnose cancer recurrences much earlier than standard surveillance methods [7,8,9,10,31,48]. In these studies, detectable ctDNA in peripheral blood during surveillance was associated with cancer relapse in almost all patients, and more importantly, ctDNA detection preceded radiologic relapse by a median time interval ranging from 3 to 11.5 months (Table 1). These data support the view that patients with detectable ctDNA during surveillance should undergo radiologic studies more frequently than standard to detect radiologic relapse earlier, which might allow curative-intent therapy in larger proportions of patients. However, the value of ctDNA-guided surveillance in colon cancer needs validation in prospective randomized studies, many of which are currently underway (Table 2).

Serum CEA, the only currently recommended blood marker for CRC surveillance, has limited utility because it lacks sensitivity and specificity [59]. In the studies discussed above, the sensitivity of serial ctDNA monitoring to predict recurrence was compared with serial CEA estimations. In the study with stage II patients, ctDNA was more frequently positive than CEA elevation at the time of radiologic recurrence (85% vs. 41%; *p* = 0.002), and ctDNA detection preceded cancer recurrence diagnosed by imaging studies by a median of 5.5 months, significantly earlier than the median 2 months of lead time observed with the serial estimation of CEA (*p* = 0.04) [10]. In the study with stage III patients, elevated CEA level following surgery or chemotherapy was associated with an inferior relapse-free interval (HR after surgery, 3.4 [95% CI, 1.5–50; *p*  = 0.02]); HR after chemotherapy, 3.05 [95% CI, 1.4–21.0; *p*  =  0.01]) [9]. However, of the 12 patients with an elevated CEA level post-ACT, 6 had detectable ctDNA, and 5 of these patients (83%) had recurrence. Of the other 6 patients with an elevated CEA level but undetectable ctDNA, only 1 (17%) had a recurrence. In the study by Reinert et al. [7], serial CEA analysis identified relapse with a sensitivity of 69% and specificity of 64% as opposed to the sensitivity of 88% and specificity of 98% with serial ctDNA measurements. In multivariable analysis, CEA elevation was not significantly associated with relapse-free survival. In this study, the mean lead time from ctDNA detection to tumor recurrence diagnosed by imaging studies was 8.7 months (range, 0.8–16.5 months; *p* < 0.001); by contrast, CEA elevation had no significant lead time. These study results suggest that serial measurements of ctDNA might be a superior tool for surveillance than CEA. Several trials, such as IMPROVE- IT2 (NCT04084249), are ongoing to define an optimal combination of ctDNA and radiologic studies for the detection of cancer recurrence, which will likely help establish evidence-based surveillance strategies.

## 7. Limitations of ctDNA

A fair evaluation of any clinical decision aiding assay is to precisely know its limitations to avoid potentially harmful decisions. Despite encouraging preliminary data, there are several barriers to wide clinical implementation of ctDNA-based testing in guiding adjuvant therapy decisions and tumor surveillance. Limited sensitivity of the ctDNA assays is an important concern, especially in the context of resected early-stage colon cancer patients where the ctDNA levels in plasma are quite low [26,31]. In the series reported by Tie et al. [10] and Reinert et al. [7], the calculated sensitivity of a single ctDNA measurement in the immediate postoperative period was modest, at around 50%. Larger sample volume [60], fragment size analysis [24], tracking multiple mutations [27], serial testing [7,8,10,61], adopting NGS panels that enable testing for a large number of genomic and epigenetic alterations [47] might improve assay sensitivity. For example, Parikh et al. have extended the assay to include epigenomic alterations, which has shown improved sensitivity (sensitivity for recurrence within one year of surgery improved from 56% to 69%) Xie et al. showed that a single plasma methylated DNA marker could detect recurrent colorectal cancer with 88% sensitivity and 95% specificity [62]. ctDNA assays must consistently detect mutations in plasma at allele fractions of < 0.1% and, to achieve that goal, should incorporate emerging techniques such as fragment size analysis, multi-UMIs to minimize PCR errors and background polishing [11,24,29,36].

DNA fragments arising from the non-neoplastic hematopoietic stem cells, or CHIP, can confound ctDNA detection leading to false-positive results [40]. The prevalence of CHIP has been reported to be 20% to 95% in healthy adults aged 60–70 years, typically at a VAF < 0.1% [63,64]. CHIP mutations generally involve genes implicated in hematologic cancer, but can affect CRC associated genes, such as TP53 and KRAS, contributing to false-positive results [65,66]. False-positive results related to CHIP, however, can be mitigated by using advanced bioinformatics filters or by matching the ctDNA sequencing with that of leukocytes [67] and/or matched tumor [7] tissues, although the optimum method remains to be elucidated.

Another barrier to overcome is the lack of standardization among different ctDNA assay methods, which significantly limits the interpretation of reported data. Discordance ctDNA results likely arise from a variety of factors, including sample collection time points, sample collection procedure, storage methods, variability of mutations queried, differences in library preparation techniques, UMIs, variant calling, and targeted error correction. Standardization of ctDNA collection, storage, and analysis methods would be critical to facilitate the wide adoption of ctDNA technology in routine clinical practice.

Although studies reported thus far have provided compelling evidence supporting the value of ctDNA in the management of patients with resected early-stage colon cancer, these studies included a small number of patients, lacked validation cohorts, and it is unknown if clearance of ctDNA with ACT is a reliable surrogate marker of survival. Ongoing studies will likely provide a definitive guidance in this scenario. A list of ongoing clinical trials to validate the ctDNA-guided adjuvant therapy strategies is provided in Table 2.

## 8. Future Perspective and Conclusions

The overarching promise of ctDNA technology in the treatment paradigm of early-stage colon cancer lies in its potential to detect MRD following resection of the primary tumor, allowing precise risk-stratification and ctDNA-guided adjuvant therapy. If this promise is fulfilled, several important objectives will be achieved: (1) adjuvant therapy can be omitted in a large number of ctDNA negative patients considered high-risk and treated with ACT by the current criteria and (2) clearance of ctDNA can be used as an endpoint in adjuvant trials evaluating novel therapies and treatment escalation/de-escalation.

ctDNA technology, once validated as a reliable surrogate marker of MRD, can potentially revolutionize the way adjuvant trials are designed and conducted. Currently, adjuvant trials are designed with the primary endpoint of either DFS or overall survival (OS), which mandate long follow-up periods. Furthermore, the ability of the existing tools to identify patients at risk of recurrence after surgery is limited, and consequently, clinical trials require enrolling a large number of patients to show a benefit. These factors have been the primary barriers to rapid progress in the field of colon cancer adjuvant therapy for many decades. ctDNA technology brings a unique opportunity of evaluating adjuvant therapies in ctDNA enriched population where clearance of ctDNA can be used as a primary endpoint (replacing DFS or OS), allowing smaller sample sizes and shorter follow up periods. Thus, ctDNA technology holds great potential in accelerating adjuvant therapy development.

Long-term toxicities, particularly neurotoxicity, are special concerns in patients who receive oxaliplatin-based ACT [68]. Although IDEA analysis has established 3 months of ACT as standard for a significant proportion of stage III patients, a sizeable fraction of patients will continue to require 6 months of therapy [69]. Clinical trials designed to inform adjuvant treatment duration based on ctDNA clearance would be of interest that may help reduce undue toxicity. Large-scale studies evaluating kinetics of ctDNA during adjuvant therapy are needed to establish clearance of ctDNA as a valid surrogate marker of survival. To that end, several large trials, such as COBRA (*n* = 1408; NCT04068103), TRACC (*n* = 1000; NCT04050345), BESPOKE (*n* = 1000, NCT04264702), and ADNCirc (*n* = 473; NCT02813928) are currently underway.

A variety of clinical trials are evaluating the clinical validity of various ctDNA-guided adjuvant therapy strategies in patients with resected early-stage colon cancer (Table 2). In the cohort of patients where the benefit of adjuvant therapy is uncertain (for example, patients with resected stage II colon cancer), randomization between standard therapy and ctDNA-guided adjuvant therapy with a primary endpoint of DFS is a logical trial design which is being currently explored in several studies (e.g., COBRA, DYNAMIC-II, and CIRCULATE). In another trial design, novel therapies are being tested in patients who continue to have detectable ctDNA following standard adjuvant therapy (e.g., NCT03803553), prompted by the compelling preliminary data showing a very high risk of cancer recurrence in this population. Another ongoing randomized phase II/III study (DYNAMIC-III) is evaluating the clinical utility of chemotherapy de-escalation or escalation informed by ctDNA status. Furthermore, a strategy of ctDNA-guided adjuvant immunotherapy (pembrolizumab) vs. placebo is being tested in patients with microsatellite instability-high solid tumors with persistent ctDNA despite curative surgery and completion of standard perioperative and/or adjuvant therapy (NCT03832569). It is important to emphasize that withholding adjuvant therapy based on a negative ctDNA result in patient population known to derive significant survival benefit with adjuvant therapy (e.g., stage III patients) is not advisable at this time, as current ctDNA assays are not well standardized and have a wide range of sensitivity.

Standardization of the ctDNA testing procedure, including the pre-analytical variables, assay characteristics, and bioinformatic analysis, is an essential focus of ongoing research. In a quality assessment study, ctDNA samples were sent to 32 different laboratories for mutation analysis, where six different cfDNA extraction methods and five different analysis methods were utilized. The study result showed an error rate that would have implications for clinical decision-making [70]. Recently the Colon and Rectal–Anal Task Forces of the United States National Cancer Institute has published a whitepaper providing detailed guidance in standardization and efficient development of the ctDNA technology [71].

ctDNA assays may provide valuable information on the genomic repertoire of the residual tumor clones. Genomic information derived from ctDNA can potentially guide targeted therapies in the adjuvant setting directed at the actionable mutations present in the residual clones that may differ from the original primary tumor due to intra-tumoral heterogeneity and/or clonal evolution.

For example, the identification of patients with residual clones harboring BRAF^V600E^ mutation opens up an opportunity of treating these patients with BRAF directed targeted therapy in adjuvant setting given the encouraging efficacy data in metastatic CRC [72]. Reinert et al. reported presence of at least one actionable mutation in the ctDNA of 9 of the 11 (81.8%) patients who had cancer recurrence detected by ctDNA alone (in the absence of radiologic relapse) [7]. As an extension of this idea, longitudinal analysis of ctDNA during adjuvant therapy may inform on mechanisms of response and resistance, providing an opportunity of genomics guided modification of therapy before overt disease progression.

ctDNA may be of great value for monitoring tumor response in the neoadjuvant setting, an evolving treatment strategy for early-stage colon cancer patients [73]. The utility of ctDNA in monitoring tumor response in neoadjuvant setting has been evaluated in a study with 34 patients of CRC and liver metastasis, in which decrease in ctDNA level after one cycle of chemotherapy was predictive of adequate tumor response suitable for tumor resection with sensitivities ranging from 82% to 91% and specificities ranging from 95% to 100% [74]. The ability of ctDNA to assess therapy response in the neoadjuvant setting is another area of future exploration, which may allow the development of novel adaptive therapy designs.

## Figures and Tables

**Figure 1 cancers-12-02808-f001:**
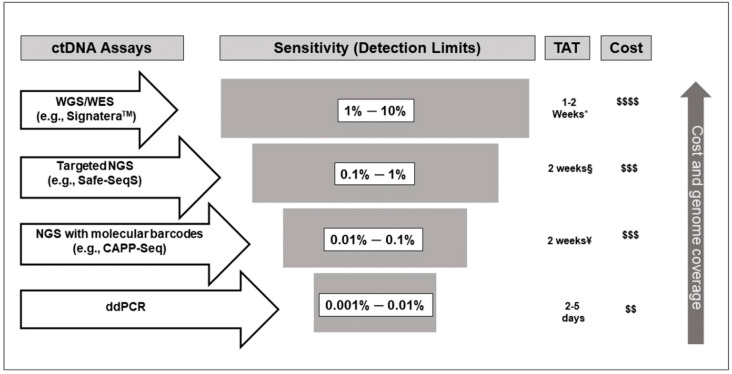
Schematic representation of sensitivity, turnaround time (TAT), cost, and genomic coverage of different circulating tumor DNA (ctDNA) assay methods illustrating variation of detection limits based on the techniques used. WGS, whole-genome sequencing; WES, whole-exome sequencing; NGS, next-generation sequencing; Safe-SeqS, safe sequencing system; CAPP-Seq, cancer personalized profiling by deep sequencing; and ddPCR, droplet digital polymerase chain reaction. * Personalized assay development takes approximately 4 weeks. After assay development, measurement of plasma ctDNA level takes 1–2 weeks. Information obtained from www.natera.com (accessed on 12 September 2020). § Information regarding TAT refers to Safe-SeqS assay for ctDNA level measurement after the personalized assay is designed (information obtained through personal communication with Dr. Bert Vogelstein [Johns Hopkins Medical Institutions]). ¥ Information regarding TAT refers to CAPP-Seq assay (information obtained through personal communication with Dr. Ash Alizadeh [Stanford Cancer Institute]). Increasing number of $ signs indicate increasing cost of the assay.

**Figure 2 cancers-12-02808-f002:**
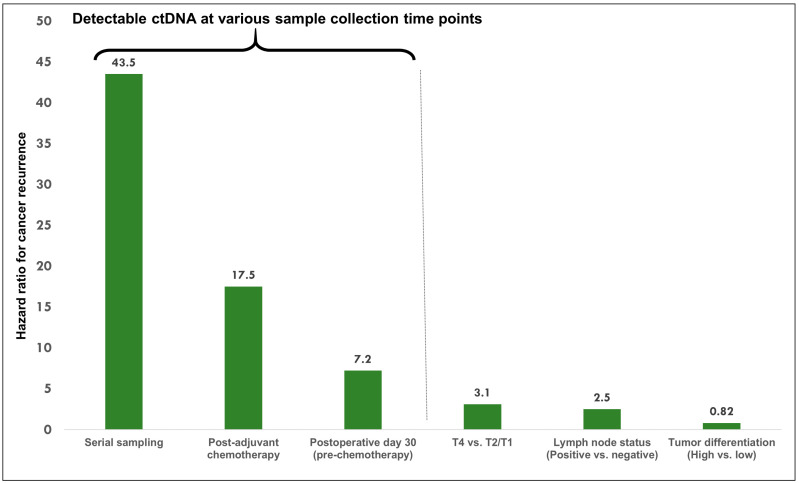
Hazard ratio (HR) for cancer recurrence with ctDNA detectable at various time points after surgery (postoperative day 30, post-adjuvant chemotherapy and serial samplings during surveillance) [7] in patients with early-stage colon cancer in comparison to other clinicopathologic risk factors-tumor extent (T4 vs. T2/T1), regional lymph node involvement status and tumor differentiation [51].

**Table 1 cancers-12-02808-t001:** Major studies supporting the clinical utility of ctDNA for minimal residual disease detection and post-therapy tumor surveillance in patients with resected early-stage colon cancer.

Study	Patient Population	ctDNA Assay Utilized	Blood Sample Collection Time Points	Major Findings	Other Relevant Findings
Tie et al. 2016 [10]	Stage II CC, *n* = 230	Tumor-informed Safe-SeqS *	4–10 weeks postop and every 3 months for up to 2 years	Cohort not receiving ACT. If positive ctDNA postop: 3-year RFS 0% (vs. 90% if ctDNA negative) and HR for recurrence 18 (95% CI, 7.9 to 40; *p* < 0.001). Cohort receiving ACT. If positive ctDNA post-ACT: HR for recurrence 11 (95% CI, 1.8 to 68; *p* = 0.001).	Median time interval between ctDNA detection and radiologic recurrence: 5.5 months (*p* = 0.001).
Scholer et al. 2017 [31]	Resected CRC, *n* = 45 (stages I to III, *n* = 21)	Tumor-informed ddPCR-based assay	Pre-op, days 8, 30, and every 3 months until month 36	Localized CRC cohort: If positive ctDNA postop: HR for recurrence 37.7 (95% CI, 4.2–335.5; *p* < 0.001). 3-year RFS 0% vs. 73%.	ctDNA detection at 3 months after surgery predicted recurrence with an average lead time of 9.4 months compared to CT scans.
Diehn et al. 2017 [27]	Stages II and III CC, *n* = 145	Tumor-informed CAPP-Seq	Single postop sample	Positive postop ctDNA: 2-year RFS 17% vs. 88% and HR for recurrence 10.3 (95% CI 2.3-46.9; *p* < 0.00001).	Monitoring multiple genomic alterations in the plasma improved sensitivity.
Reinert et al. 2019 [7]	Stages I to III CRC, *n* = 130	Tumor-informed, personalized, multiplex, PCR–based NGS **	Preop, postop day 30, and every 3 months for up to 3 years.	HR for recurrence with positive ctDNA: Postop day 30: 7.2 (95% CI, 2.7-19.0; *p* < 0.001). Shortly after completion of ACT: 17.5 (95% CI, 5.4–56.5; *p* < 0.001). Serial monitoring post-ACT: 43.5 (95% CI, 9.8–193.5, *p* < 0.001).	Serial ctDNA analyses revealed disease recurrence up to 16.5 months ahead of radiologic imaging (mean, 8.7 months; range, 0.8–16.5 months).
Tie et al. 2019 [9]	Stage III CC, *n* = 96	Tumor-informed Safe-SeqS *	4–10 weeks postop and within 6 weeks of ACT completion	HR for recurrence with positive ctDNA: Postop: 7.5 on multivariable analysis (95% CI, 3.5–16.1; *p* < 0.001). Shortly after ACT: 6.8 (95% CI, 11.0-157.0; *p* < 0.001).	RFS at 3 years in patients with ctDNA positive vs. negative: Postop 47% vs. 76%, post-ACT 30% vs. 77%.
Tarazona et al. 2019 [8]	Stages I to III CC, *n* = 150	Tumor-informed ddPCR	Preop, 6–8 weeks postop and every 4 months up to 5 years.	HR for recurrence with positive ctDNA: Postop (after multivariable adjustment):11.6 (95% CI, 3.6–36.8; *p* < 0.001). Post- ACT: 10.02 (95% CI, 9.2–307.3; *p* < 0.0001).	Positive ctDNA during surveillance preceded radiological recurrence with a median lead time of 11.5-months.
Parikh et al. 2019 [47] ^¥^	Stages 0-IV CRC, *n* = 72. (Stage IV, *n* = 24)	Tumor-uninformed assay (LUNAR-1)	Postop and post-ACT	Postop positive ctDNA: HR for recurrence 8.7 (*p* < 0.0001), PPV 100%, NPV 76%. Post-ACT positive ctDNA: HR for recurrence 9.3 (*p* < 0.0001), PPV 100%, NPV 76%	Detection of ctDNA within 1 year of surgery predicted recurrence with a sensitivity of 69% and specificity of 94%
Wang et al. 2019 [48]	Stages I to III CC, *n* = 58	Tumor-informed Safe-SeqS *	Postop at 1 month and then every 3–6 months	Recurrence rate among patients with positive ctDNA at any time point after surgery was 77% (10 of 13 patients). None of the 45 patients with negative ctDNA throughout the follow-up experienced a relapse (median follow-up 49 months).	Positive ctDNA preceded radiologic and clinical evidence of recurrence by a median of 3 months.

Abbreviations: CC, Colon cancer; n, number of patients; Preop, preoperative; Postop, postoperative; Post-ACT, after adjuvant chemotherapy; ctDNA, circulating tumor DNA; Safe-SeqS, safe sequencing system; HR, hazard ratio; CI, confidence interval; MRD, minimal residual disease; CRC, colorectal cancer; PPV, positive predictive value; NPV, negative predictive value; CAPP-Seq, cancer personalized profiling by deep sequencing; RFS, relapse-free survival; ddPCR, droplet digital polymerase chain reaction; and NGS, next-generation sequencing. * One somatic mutation was tracked in the plasma samples of each patient using a panel of 15 genes that are commonly mutated in CRC. ** Sixteen high-ranked patient-specific somatic single-nucleotide variants and short indels were selected for each patient by tumor whole-exome sequencing. Plasma samples with at least 2 variants detected were defined as ctDNA positive. ^¥^ Data collected from the poster (abstract #3602) published in the 2019 ASCO annual meeting [47].

**Table 2 cancers-12-02808-t002:** Ongoing and upcoming clinical trials investigating the clinical utility of ctDNA in the management of patients with early-stage colon cancer *.

Study Identifier (Acronym)	Study Design	Study Population	Target Patient Number	ctDNA Assay Utilized	Study Description/Primary Endpoint
NCT04068103 (NRG GI005, COBRA)	Phase II/III	Resected stage IIA patients without high-risk features	1408	LUNAR-1 (Guardant Health)	Arm 1: Active surveillance. Arm 2: ctDNA directed therapy (ctDNA positive→mFOLFOX6/CAPOX for 6 months, ctDNA negative→active surveillance). The primary endpoints: Clearance of ctDNA with ACT (phase II) and RFS (phase III)
NCT04120701 (CIRCULATE)	Phase III	Resected Stage II patients	1980	Not reported	ctDNA positive→randomized (2:1) to receive ACT or no ACT. ctDNA negative→surveillance. Primary endpoint: 3-year DFS in ctDNA positive patients randomized to ACT or to follow-up.
ACTRN12615000381583 (DYNAMIC-II)	Phase III	Resected stage II patients	450	Safe-SeqS	Arm A: positive ctDNA→ACT, negative ctDNA→surveillance. Arm B: Treated at the discretion of the clinicians. Primary outcome measures-RFS and to evaluate whether a ctDNA guided adjuvant therapy strategy affects the number of patients treated with ACT.
ACTRN12617001566325 (DYNAMIC-III)	Phase II/III	Resected stage III patients	1000	Safe-SeqS	Arm A: Standard of care. Arm B: ctDNA informed (ctDNA negative→therapy de-escalation; ctDNA positive→therapy escalation). Primary endpoint: 3-year RFS (to demonstrate that a therapy de-escalation/escalation strategy informed by ctDNA is non-inferior to standard of care treatment).
NCT04084249 (IMPROVE-IT2)	Phase III	Resected high-risk stage II and stage III patients	254	Not reported	Randomization between ctDNA-guided surveillance and standard surveillance. Primary endpoint- Fraction of patients with relapse receiving curative-intent treatment
NCT03803553	Phase III	Resected stage III patients	500	LUNAR-1 (Guardant Health)	Patients are enrolled after standard adjuvant chemotherapy in one of the 2 arms: 1. ctDNA negative: Surveillance; 2. ctDNA positive: (a) MSS patients→6 months of FOLFIRI vs. surveillance, (b) MSI high→6 months of nivolumab, (c) BRAF mutant/MSS→6 months of BRAF directed therapy. Primary outcome measures: 5-year DFS and clearance rate of ctDNA.
NCT04050345(TRACC)	Prospective, observational	Stage I, II and III patients with CRC	1000	Not reported	Multicenter, prospective study involving the collection and analysis of tumor tissue, serial blood samples, and clinical data in patients with newly diagnosed stage I, II and III CRC. Primary outcome measures: 1. The incidence of detectable ctDNA in patients with stage II and III CRC pre-operatively, 2. The correlation between detectable ctDNA at the first postoperative visit and DFS.
NCT04264702 (BESPOKE)	Prospective Observational study	Resected stage II and III colon cancer	1000	SIGNATERA™	Patients will undergo a ctDNA testing following surgery and may be recommended for adjuvant chemotherapy or observation by their treating clinician. Follow up period- up to 2 years with periodic blood sample collection for ctDNA assay. Control arm will consist of matched stage II or stage III patients who have a minimum of 2 years of clinical follow-up data. Primary outcome measures: 1. To examine the impact of ctDNA on adjuvant treatment decisions. 2. To determine the rate of tumor recurrence while asymptomatic using ctDNA.
NCT04259944 (PEGASUS)	Phase II	Resected MSS stage III and high-risk stage II (T4N0) patients	140	LUNAR-1 (Guardant Health)	ctDNA guided ACT. (i) ctDNA positive→CAPOX for 3 months; (ii) ctDNA negative→capecitabine for 6 months but will be retested after 1 cycle, and if found ctDNA positive, will be switched to CAPOX. Primary outcome measure: Number of post-surgery and post-adjuvant false-negative cases after a double ctDNA-negative detection.

Abbreviations: ctDNA, circulating tumor DNA; mFOLFOX6, 5-fluorouracil, leucovorin, and oxaliplatin; RFS, relapse-free survival; CAPOX, capecitabine and oxaliplatin; DFS, disease-free survival; ACT, sdjuvant chemotherapy; MSS, microsatellite stable; MSI, microsatellite instability; Safe-SeqS, safe sequencing system; FOLFIRI, 5-fluorouracil, leucovorin, and oxaliplatin; and CRC, colorectal cancer. * Clinicaltrials.gov accessed between 8 August 2020 and 18 August 2020.

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
