# Peer review of "The Promise of Circulating Tumor DNA (ctDNA) in the Management of Early-Stage Colon Cancer: A Critical Review"

_cancers, 2020, doi:10.3390/cancers12102808_

Round 1

Reviewer 1 Report

This review faces all the salient points regarding the potential utility of cell free tumor DNA in Early-Stage colon cancer providing information from a range of perspectives. In particular, they reported in a very clear and complete manner, the actual knowledge on the potential role of cfDNA as surrogate marker of MRD, as prognostic indicator, and as a tool to monitor efficacy of adjuvant therapy.

The list of studies supporting the clinical utility of cfDNA as well as the ongoing clinical trials to validate the ctDNA-guided adjuvant therapy strategies provided in the paper are very appropriate and salient points are well illustrated in the tables.

They also dedicated a robust session on the actual limitation of cfDNA emphasizing that there are still several barriers to wide clinical implementation of ctDNA-based testing in guiding adjuvant therapy decisions and tumor surveillance. In addition, they underlined that the overarching promise of ctDNA technology in the treatment paradigm of early-stage colon cancer lies in its potential to detect MRD following resection of the primary tumor, allowing precise risk-stratification and ctDNA-guided adjuvant therapy. They conclude that if this promise is fulfilled, liquid-biopsy can potentially revolutionize the way adjuvant trials are designed and conducted.

The manuscript is well and clearly written and could be published in the journal in this form after minor revision.

The authors may have missed a recent paper (Allegretti et al., 2020) on early CRC that is relevant to most issues discussed in their review. Therefore, I strongly suggest adding this reference.

Allegretti M, Cottone G, Carboni F, Cotroneo E, Casini B, Giordani E, Amoreo CA, Buglioni S, Diodoro M, Pescarmona E, Zazza S, Federici O, Zeuli M, Conti L, Cigliana G, Fiorentino F, Valle M, Giacomini P, Spinella F. Cross-sectional analysis of circulating tumor DNA in primary colorectal cancer at surgery and during post-surgery follow-up by liquid biopsy. J Exp Clin Cancer Res. 2020 Apr 20;39(1):69

Author Response

Dear Reviewer 1:

On behalf of my coauthors, I am pleased to submit the response to your critique and the revised manuscript entitled, “The Promise of Circulating Tumor DNA (ctDNA) in the Management of Early-Stage Colon Cancer: A Critical Review.” We much appreciate the critique. We want to express our most sincere gratitude to you and the editorial team for a thorough review. We have made a sincere effort to address the critique in our revised manuscript that, we believe, will enhance the value and the clarity of this article.

Critique:

The authors may have missed a recent paper (Allegretti et al., 2020) on early CRC that is relevant to most issues discussed in their review. Therefore, I strongly suggest adding this reference.

Allegretti M, Cottone G, Carboni F, Cotroneo E, Casini B, Giordani E, Amoreo CA, Buglioni S, Diodoro M, Pescarmona E, Zazza S, Federici O, Zeuli M, Conti L, Cigliana G, Fiorentino F, Valle M, Giacomini P, Spinella F. Cross-sectional analysis of circulating tumor DNA in primary colorectal cancer at surgery and during post-surgery follow-up by liquid biopsy. J Exp Clin Cancer Res. 2020 Apr 20;39(1):69.

Response

We have incorporated this relevant study in our manuscript by adding the following lines in section 4:

‘A similar study by Allegretti et al.[50] reported that persistence and absence of ctDNA at the time of first post-operative (3 month) follow-up were associated with fast relapse and a disease-free status in 3 and 7 patients, respectively. In addition, this study reported improved sensitivity( 58.8% to 63.6%) when ctDNA result was analyzed in combination with serum CEA level.’

I have attached the revised manuscript as well.

We look forward to receiving your response.

Best regards,

Sakti Chakrabarti

Reviewer 2 Report

This review by Chakrabarti and colleagues represents a complete overview of published studies evaluating clinical utility of ctDNA in the management of patients with early-stage colon cancer and its potential to guide further adjuvant therapy strategies. After listing a few key aspects of the ctDNA biology, the authors also discuss the clinical utility of ctDNA in the assessment of adjuvant chemotherapy efficacy, cancer surveillance and recurrence. Ultimately, they describe the rationale and design of the ongoing clinical trials involving ctDNA that investigate its validity to guide treatment strategies.
Overall this review is clearly written, and nicely summarizes the main publications on the topic. I have a few major and minor suggestions to improve this manuscript prior to publication in this journal:

1) Paragraph 2, the title “ The Biology of ctDNA Reveals the Ability of Tumors to Send Novel Messages that Can Be Intercepted in Transit” is not clear at all. I would suggest rephrasing that.
2) Paragraph 2 should be shortened, as the basic biology of ctDNA has been largely described in other reviews, and it’s really not the main focus of the current one.
3) The authors should include citations of studies that showed how ctDNA can be isolated and exploited from cerebrospinal fluid, in addition to stool, urine and saliva. Bronchial washings could be added as well.
4) Figure 1 illustrates the sensitivity levels of common ctDNA assays. However, it does not include any data about the different TAT and costs of each assay, which I think may be useful to the readers.
5) I think the authors should discuss the role of ctDNA in identifying MRD in patients treated with immunotherapy (i.e. lung cancer patients).
6) ddPCR stands for “droplet digital PCR” and not “digital droplet PCR”. Please fix this.
7) There is a typo “ctDAN” on page 7 line 177. Please fix this.
8) There is a typo “alternations” on page 10 line 309.
9) References #47 and #61 actually refers to the same study.

Author Response

Dear Reviewer 2:

On behalf of my coauthors, I am pleased to submit the response to your critique and the revised manuscript entitled, “The Promise of Circulating Tumor DNA (ctDNA) in the Management of Early-Stage Colon Cancer: A Critical Review.” We much appreciate the critique. We want to express our most sincere gratitude to you and the editorial team for a thorough review. We have made a sincere effort to address the critique in our revised manuscript that, we believe, will enhance the value and the clarity of this article.

Critique:

1) Paragraph 2, the title “ The Biology of ctDNA Reveals the Ability of Tumors to Send Novel Messages that Can Be Intercepted in Transit” is not clear at all. I would suggest rephrasing that.

Response

We have changed it to ‘The Biology of ctDNA’.

2) Paragraph 2 should be shortened, as the basic biology of ctDNA has been largely described in other reviews, and it’s really not the main focus of the current one.

Response

We agree that the biology of ctDNA is not an essential component of this review. We elaborated on the ctDNA biology keeping the clinicians in mind who may not have adequate familiarity with the basic biology of ctDNA, given that ctDNA technology is new and not used widely in the clinics. We want to request the reviewer and the editorial team to keep this paragraph in its current form. However, if the reviewer and the editorial team strongly feel that this section should be shortened, we would modify this section as follows-

 ctDNA is single- or double-stranded DNA fragments released from neoplastic cells, which typically constitute less than 1% of the total cfDNA[11]. Pioneering investigation by Leon et al. back in the 1970s  led to the realization that the rapid cell turnover rate in malignant tumors results in an increased concentration of cfDNA in the blood of cancer patients compared to healthy individuals [12], which was subsequently confirmed by other investigators[13]. Noteworthy, however, several conditions unrelated to cancer, such as acute trauma, ischemia, infection, or inflammation, can increase cfDNA concentrations in the circulation[14,15]. ctDNA is released into the circulation predominantly by apoptosis[16], necrosis, phagocytosis, and carried by exosomes [17,18]. ctDNA can also be detected in non-blood body fluids, such as urine, saliva, sputum, stool, pleural fluid, and cerebrospinal fluid (CSF)[19]. These DNA fragments are continuously released by neoplastic cells, undergo rapid degradation by blood nucleases, and are finally cleared by the liver and kidneys[20], accounting for their short half-life in circulation (16 minutes to 2.5 hours)[21]. The rapid turnover of ctDNA in circulation makes it an attractive target for obtaining a real-time account of mutation dynamics and tumor burden[22,23]. Additionally, DNA fragments derived from cancer cells are typically shorter in length, which forms the basis for selecting fragments between 90 bp and 150 bp to improve the detection sensitivity of ctDNA assays[24]. Notably, plasma samples are preferable to serum samples for ctDNA analysis as the latter contain larger quantities of DNA from leukocytes lysed during the clotting process, and thereby, increasing the background versus signals ratio and interfering with the assay[25].

3) The authors should include citations of studies that showed how ctDNA can be isolated and exploited from cerebrospinal fluid, in addition to stool, urine and saliva. Bronchial washings could be added as well.

Response

Thank you for this suggestion. We have added the following line in section 2 along with the reference-

‘ctDNA can also be detected in non-blood body fluids, such as urine, saliva, sputum, stool, pleural fluid, and cerebrospinal fluid (CSF).’

4) Figure 1 illustrates the sensitivity levels of common ctDNA assays. However, it does not include any data about the different TAT and costs of each assay, which I think may be useful to the readers.

Response

We highly appreciate this suggestion. We have incorporated those informations in figure 1.

5) I think the authors should discuss the role of ctDNA in identifying MRD in patients treated with immunotherapy (i.e., lung cancer patients).

Response

We appreciate this critique. We agree that the role of immunotherapy in the adjuvant treatment of early-stage colon cancer is evolving and needs to be included in the discussion. To address this critique, we have added the following lines in the future perspective section (section 8) –

 ‘Furthermore, a strategy of ctDNA-guided adjuvant immunotherapy with pembrolizumab vs. placebo is currently being tested in a clinical trial in patients with microsatellite instability-high solid tumors with persistent ctDNA despite curative surgery and completion of standard perioperative and/or adjuvant therapy (NCT03832569).’ 

6) ddPCR stands for “droplet digital PCR” and not “digital droplet PCR”. Please fix this.

Response

We apologize for this oversight. We have fixed it in the manuscript.

7) There is a typo “ctDAN” on page 7 line 177. Please fix this.

Response

We have fixed this typo in the manuscript.

8) There is a typo “alternations” on page 10 line 309.

Response

We have fixed this typo in the manuscript.

9) References #47 and #61 actually refers to the same study.

Response

This reference issue has been fixed.

I have attached the revised manuscript as well.

We look forward to receiving your response.

Best regards,

Sakti Chakrabarti

Round 2

Reviewer 1 Report

The review in this form is fine to me. 

Reviewer 2 Report

I thank the authors for addressing my concern. I still believe the first paragraph should be shortened as suggested by the authors in the point by point response to reviewers. I recommend this review for publication.